# Health Literacy Levels among Italian Students: Monitoring and Promotion at School

**DOI:** 10.3390/ijerph18199943

**Published:** 2021-09-22

**Authors:** Veronica Velasco, Andrea Gragnano, Luca Piero Vecchio

**Affiliations:** 1Psychology Department, Milano-Bicocca University, 20126 Milan, Italy; andrea.gragnano@unimib.it (A.G.); luca.vecchio@unimib.it (L.P.V.); 2Regione Lombardia, DG Welfare, 20124 Milan, Italy; promozionesalute@regione.lombardia.it

**Keywords:** health literacy, adolescent, health promotion, school health, school-aged children, measurement, family support, school support

## Abstract

Health literacy was identified as an important determinant of health, particularly for adolescents. However, more efforts are needed to monitor this construct and provide inputs for policy development. This study aims to: (a) Assess the validity and reliability of the Italian version of the Health Literacy for School-Aged Children (HLSAC-I); (b) Identify the health literacy levels among Italian students and compare them with other countries’ levels; and (c) Identify the associations between health literacy and multiple social determinants (social stratifiers, family, and school connectedness). Data came from the Health Behaviour School-Aged Children survey, carried out in the Lombardy region in northern Italy in 2018. A representative sample of 2,287 13- and 15-year-old Lombardian students was involved. The results support the validity and reliability of the HLSAC-I. A total of 18.7% of the sample reported low levels, and only 6.8% reported high levels. Italian students reported the lowest levels of health literacy compared with other countries. School connectedness and educational approach are the most relevant associated factors. This study confirms a school’s role in reducing inequalities and promoting health. It highlights the importance of monitoring health literacy and implementing health promotion policies at school through a whole-school approach.

## 1. Introduction

In recent years, the World Health Organization (WHO) and other international bodies have emphasized health literacy as an important determinant of health [1,2,3,4]. Research in this area has increased to expand knowledge of the effects of health literacy, the elements that can promote it, and the most effective strategies to improve it [5]. There are many definitions of health literacy in the existing literature [6], but the WHO Glossary of Health Promotion [7] defines this construct as “the cognitive and social skills which determine the motivation and ability of individuals to gain access to, understand and use information in ways which promote and maintain good health”. Moreover, different levels of health literacy can be identified according to their practical applications in everyday life. Basic or functional health literacy includes the literacy, knowledge, and other skills sufficient to acquire and act on information about health risks and that health services use; it refers to the skills needed to function effectively in everyday life. Interactive health literacy refers to the skills useful to extrapolate, understand, and distinguish health information through different sources and to apply newly learned information in different situations. Critical health literacy describes the more developed cognitive and social skills that make a person able to critically analyse health-related information that is drawn from multiple sources and to use it to exert greater control over both one’s own health decisions and the external influences on those decisions. Critical health literacy includes a healthy citizenship that empowers people to join social and political processes and to modify the determinants of health [1,8,9,10].

Several studies have shown a significant relationship between health literacy and healthy behaviours [11,12,13,14], the use of and access to health and social services, health conditions and health outcomes, the ability to manage long-term health conditions, inequalities [15,16,17], and social capital [18]. Moreover, health literacy supports empowerment, participation, and autonomous development [8]. During the current emergency phase related to COVID-19, health literacy has become an even more important construct [19,20]. Health literacy increases the likelihood of being well-informed and aware of risks, resources, and recommendations; it may encourage the adoption of supportive and protective behaviours to achieve public health and serve as a protective factor against inequities. Moreover, it represents a key resource for managing rapid changes in epidemiological situations and any subsequent indications for countering infections, and it is a key resource for preventing an “infodemic”, i.e., a global epidemic of misinformation that spreads rapidly through social media and other platforms [21,22].

A specific focus on adolescent health literacy is needed. Childhood and adolescence are among the most effective times to intervene and promote health skills. Health literacy development starts in childhood [23,24,25], and adolescence is a crucial moment for promoting healthy development and well-being in adulthood [26,27]. Moreover, students’ relationship with school creates an optimal context where it is possible to promote health literacy [18,28,29,30,31,32,33]. A school typically provides universal access and has a strong impact on inequities. Several studies have demonstrated a strong link between health literacy, education, and learning [8]. A school can promote access to information and technology and offer health promotion programs that can facilitate increases in health literacy. Moreover, the school can provide interprofessional education and collaborative practice that has been shown effective in promoting health literacy in other contexts [34,35,36]. 

Many authors have argued that the health literacy construct needs a different conceptualization for adolescents to include their needs and the influences of their specific context [1,37,38]. Health literacy is content- and context-specific [8,18], and so it is inevitable that health literacy should be defined differently for such an ever-changing population as adolescents. In addition to cognitive dimensions, the construct should include the affective, operational or behavioural, and specific technical attributes of adolescents [1,39].

Although the importance of health literacy is widely recognized, efforts are needed to monitor health literacy in different population groups and to provide inputs for policy and intervention developments and improvements [33]. However, few comparative studies have focused on the health literacy population level in different countries, and methodological improvement is necessary. In particular, regarding adolescent health literacy assessments, more specific conceptual and operational definitions are necessary [37]. Most of the existing studies have measured only functional literacy, without considering a broader health literacy construct [40,41]. More specific development and validation processes are also needed, and more information about the psychometric properties of the scales is necessary [37,42].

A scale recently developed to assess adolescents’ health literacy is the Health Literacy for School-Aged Children scale (HLSAC) [41,43,44]. It is a brief, theory-based, and comprehensive instrument to measure subjective health literacy among school-aged children. It refers to a multidimensional conceptualisation of health literacy and includes five core components: theoretical knowledge, practical knowledge, critical thinking, self-awareness, and citizenship. The scale was validated in Finland [43] through a long and specific process, and its invariance was verified through many other countries’ versions [44,45]. The HLSAC was included in the Health Behaviour in School-Aged Children (HBSC) protocol. The HBSC is a cross-national and comprehensive survey about health behaviours, conditions, and social determinants in students who are 11, 13, and 15 years old. It is promoted by the WHO, and approximately 50 countries are involved in it. The HBSC represents a useful opportunity to monitor health literacy levels among adolescents, to compare the results among countries and over time, and to verify the relationships between health literacy, social determinants, and health behaviours [28].

Improving health literacy among adolescents and organizing health promotion policies are also necessary to understand the determinants and associated factors of health literacy. Literature reviews have shed light on the influence of sociodemographic indicators (e.g., sex), educational levels (e.g., school experience, school performance), and socioeconomic factors (e.g., family income, parents’ education level, environmental contextual factors) [1,11,15,46,47,48]. These variables were recognised as social stratifiers and factors of inequities [16,49]. The influence of the family was particularly investigated. Parental health literacy, health status, and educational variables were shown to determine adolescent health literacy levels [1,15,50]. Recent studies have highlighted the relevance of parents’ educational styles to communicating health information. Studies have shown that young people depend on their parents to access health information and resources, and, therefore, different approaches are used to share or filter health information [11,51]. As stated above, school influence was also extensively affirmed. However, few studies have investigated specific educational factors, other than education levels and school years [46]. School type, performance, and aspiration were shown to be relevant determinants [16,40,52]. However, the literature about health promotion has affirmed the importance of including health literacy in the whole-school approach of the Health Promoting Schools and of considering teachers’ roles as facilitators [18,28,31]. More investigation is needed to better understand the comprehensive effects of and the links between health literacy and adolescents’ proximal social contexts, such as family and school [53]. The associations with educational processes used in both contexts should also be considered.

This study aims to assess the validity and reliability of the Italian version of the Health Literacy for School-Aged Children (HLSAC-I). Moreover, we seek to identify the health literacy levels among Italian students and compare them with other countries’ levels. Third, we aim to identify the associations between health literacy and multiple social determinants. In particular, sociodemographic variables and social stratifiers, as well as family and school connectedness and support will be considered.

## 2. Materials and Methods

This study’s data came from the Health Behaviour in School-Aged Children (HBSC) survey, carried out in the Lombardy region in northern Italy in 2018. In Italy, the HBSC survey involves a national sample and representative regional samples for each region. For this study, the regional sample of the Lombardy region was considered.

### 2.1. Sample

A representative sample of 2287 13- and 15-year-old Lombardian students was used in this study. Students were selected via a random cluster sampling of schools, followed by a random sampling of classes within each school.

Only the students who answered the health literacy scale were considered for the analysis. The final sample included 2128 students of 130 classes (97.7% response rate): 63 classes of the third year of middle school and 67 of the second year of high school. Fifty-three percent of the participants were female, and the students were equally distributed across the two age groups considered. Table 1 reports the sample characteristics.

### 2.2. Procedure

The HBSC study conducted in Italy in 2018 was promoted by the National Health Institution in collaboration with a national academic network, the Ministry of Education and Health, and the regional government and regional school office in Lombardy.

The HBSC international study protocol was followed to define the instruments, the participant consent, anonymity, and confidentiality procedures, and data collection and processing [54]. A formal presentation of the survey was sent to all schools in the region, and the sampled schools were invited to participate. All sampled schools were contacted to promote their participation by the survey coordinators, the local school offices, and the local health units. The survey was administered during a school day by health professionals who were specially trained.

The Italian HBSC study protocol and questionnaire were formally approved at the national level by the Ethics Committee of the Italian National Institute of Health and at the regional level by the regional school office and the regional government committee. At each school, the approval of school principals, who also managed parental consent, was required. Finally, every participant was informed about the study, agreed to participate, and was guaranteed the confidentiality of his or her data. Participation was voluntary, and students could leave the survey at any time.

### 2.3. Measures

All measures were derived from the HBSC international protocol.

#### 2.3.1. Health Literacy

The Health Literacy for School-Aged Children (HLSAC) instrument was used to measure self-reported health literacy. The instrument includes 10 items and five components (theoretical knowledge, practical knowledge, critical thinking, self-awareness, and citizenship). Two items measure each component, and a Likert scale is included (1 = ‘Not at all true’; 4 = ‘Absolutely true’). The total score is calculated with the mean of the items and can be divided into three categories: ‘low’ (scores 10–25), ‘moderate’ (scores 26–35) and ‘high’ (scores 36–40). The items were translated into Italian using a translation and back-translation process. Three independent translations were involved. First, the English version of the scale was translated into Italian; then, the Italian version of the scale was translated into English. The correspondence between the original version of the scale and its back-translation was checked. A few differences emerged, and they were discussed until a consensus was reached.

#### 2.3.2. Sociodemographic Variables and Social Stratifiers

Sex, school level, mother’s and father’s educational level, and family economic condition were assessed. The school level included the two age groups considered: 13- and 15-year-olds. Five parents’ educational levels were considered (1 = primary school; 5 = degree). Economic condition was measured through the Family Affluence Scale (FAS III), an indicator of family affluence that includes six items about material resources, such as the number of bathrooms at home or cars. The indicator ranges from 0 (lowest affluence) to 13 (highest affluence). For the sample description, the indicator was recoded into low, middle, and high levels according to a threshold determined by the HBSC national and international networks [55].

#### 2.3.3. Family Variables

Quality of maternal and paternal communication and family support were assessed as indicators of education and relationships in the familial context. The quality of parental communication was assessed by asking “How easy is it for you to talk to your father/mother about things that really bother you?” on a Likert scale, from 1 (1 = very easy) to 5 (5 = don’t have or see this person). Item scores were reversed to make higher scores correspond to better communication.

The HBSC protocol was adapted the Family Support Scale from the Multidimensional Scale of Perceived Social Support about family [56]. It included 4 items (e.g., “My family is willing to help me make decisions”) on a Likert scale, from 1 (1 = very strongly disagree) to 7 (7 = very strongly agree). An overall indicator was calculated through the mean of the items. The Cronbach’s α was 0.91.

#### 2.3.4. School Variables

School satisfaction, teacher support, and school ability to promote competence and autonomy were assessed as indicators of the relationship of a school and educational processes in this context. School satisfaction was measured through a single item (“How do you feel about school at present?”) on a Likert scale, from 1 (1 = I like it a lot) to 4 (4 = I don’t like it at all). This item was an indicator of students’ emotional and psychological connectedness to their school. Item scores were reversed to make higher scores correspond to better satisfaction.

Teacher support was measured by three items (e.g., “I feel that my teachers care about me as a person”) on a Likert scale, from 1 (1 = strongly agree) to 7 (5 = strongly disagree). Item scores were reversed to make higher scores correspond to more support, and an overall indicator was calculated through the means of the items. The Cronbach’s α was 0.77.

The school-related competence and autonomy scale was also considered. It included 8 items (e.g., “My teachers guide me on how to solve tasks”) on a Likert scale, from 1 (1 = strongly agree) to 7 (5 = strongly disagree). Item scores were reversed to make higher scores correspond to greater competence support, and an overall indicator was calculated through the mean of the items. The Cronbach’s α was 0.87.

### 2.4. Analysis

Data collection took place in school classes; therefore, the clustering of health literacy scores at the school and classroom levels was examined to determine if multilevel analyses were necessary. The intraclass correlation coefficients of all items were lower than 0.05, suggesting the absence of clustered data [57]. Thus, no multi-level analyses were necessary.

Because the percentage of respondents with missing values was low (6%), listwise deletion was deemed usable [58]. Therefore, all the analyses were performed on a sample of 2148 students. To test the factor validity of the HLSAC-I, we replicated, with a confirmatory factor analysis (CFA), the one-factor model that was proposed as the final model by the validation study of the original scale [43]. Because in the same study a five-factor model was originally hypothesized and some researchers have proposed a validation of the HLSAC with a five-factor model [45], we also tested a five-factor model (i.e., theoretical knowledge, practical knowledge, critical thinking, self-awareness, and citizenship), with two items per factor.

The HLSAC response scale is a four-point Likert scale. With fewer than five answer categories, the HLSAC items cannot be considered continuous variables in a CFA and should be treated as categorical. In this situation, using the maximum likelihood (ML) estimation method may be problematic; therefore, we employed the weighted least square mean and variance adjusted (WLSMV) method, which is specifically designed for ordinal data [59]. To evaluate the number of problems derived from employing the ML estimator, instead of WLSMV, via the HLSAC-I questionnaire, we replicated the model with the ML estimator.

The goodness-of-fit of the models was assessed with the comparative fit index (CFI), the root mean square error of approximation (RMSEA), and the standardized root mean square residual (SRMR) models. Because current computations of CFI and RMSEA under WLSMV estimation produce overly optimistic results, we calculated CFIcML and RMSEAcML [60]. These indices approximate what the fit index values would have been, had the data not been categorized so that the commonly used cut-off value established under ML could be adopted [60]. Model fit was considered acceptable when CFI > 0.90, SRMR ≤ 0.08, and RMSEA ≤ 0.08 and good when CFI > 0.95, SRMR ≤ 0.07, and RMSEA ≤ 0.05 [61,62,63]. We also reported the standardized factor loadings and items R^2^. Item R^2^ indicates the proportion of item variance explained by the latent variable. Factor loadings should all be significant, and items R^2^ ≥ 0.26 [64]. Reliability was measured with omega (ω), which should be higher than 0.70 [65].

Frequencies and percentages were used to describe the sample characteristics and the health literacy levels. We also compared the 15-year Italian results with those of other countries that included their HLSAC in the HBSC 2017–18 study [41]. In particular, we selected the countries with the lowest values of health literacy. A *t*-test was run to compare the Italian and Austrian values, using the mean values, standard deviation and the sample size of the present study and the international data. Moreover, two chi-square tests were used to compare the Italian health literacy levels with the Czechian and German health literacy levels, using the observed frequencies for each country. No control variables were included in this study or in the confronting study [41].

To identify the association between health literacy and multiple social determinants, a hierarchical linear regression was run. Model 1 included the following sociodemographic characteristics and social stratifiers: sex, school level, mother’s and father’s educational level, and family economic condition. Model 2 added these family characteristics to the previous variables: quality of maternal and paternal communication, and family support. Model 3 added these school characteristics: school satisfaction, teacher support, and school-related competence and autonomy.

A hierarchical regression was chosen to verify the relative importance of the different associated factors and contexts.

The key assumptions for running the multiple linear regression analyses were tested. Homoscedasticity was tested, through a visual examination of the graph of standardized residuals, against standardized predicted values. The absence of multicollinearity was verified through the tolerance T and the variance inflation factor (VIF) < 5.00. The absence of autocorrelation was tested through the Durbin–Watson test, with values from 1.5 to 2.5, and the absence of influential observations was verified through Cook’s Distance < 1.00.

The lavaan package [66] for R [67] was used for scale validation, and IBM SPSS Statistics 27 was used for all the other analyses.

## 3. Results

### 3.1. HLSAC-Italian Version Validation

The one-factor model provided acceptable fit indices: χ2(df) = 262(35), *p* < 0.001; CFIcML = 0.92; RMSAcML = 0.08; SRMR = 0.04. The five-factor model provided equivalent fit indices, except for a slight increase in CFI cML: χ2(df) = 218(25), *p* < 0.001; CFIcML = 0.93; RMSAcML = 0.08; SRMR = 0.04. However, the covariance matrix of latent variables was not positive definite, as the correlation between latent variables was too strong; many were higher than 0.82, with some higher than 1 (Heywood case). Therefore, the one-factor model was considered a better model. Figure 1 shows standardized factor loadings and R^2^ in the one-factor model. All factor loadings were statistically significant, and R^2^ was equal to or higher than 0.26. Only the item “ability to follow the instructions given by doctors and nurses” had a low R^2^ at 0.18. The reliability of the latent factor health literacy was good, with ω = 0.80. Using ML to estimate the one-factor model resulted in slightly better fit indices (χ2(df) = 327(35), *p* < 0.001; CFI = 0.93; RMSA = 0.06; SRMR = 0.04) and equivalent model estimates.

### 3.2. Health Literacy Levels

The total sample reported a health literacy mean of 29.08, and 18.7% of the sample showed a low level of health literacy, 74.5% a moderate level, and 6.8% a high level. Table 1 shows the descriptive analysis for the total sample and subgroups according to sex, age and economic condition.

It is possible to compare the 15-year-olds’ results with those of other countries that included HLSAC in the HBSC 2017–18 study [41]. The health literacy levels of Italian students are the lowest. The Italian mean value is lower than that of any other country. In particular, when we compared the Italian mean value with the Austrian mean value, the lowest value reported in the previous study, the *t*-test was significant (t = 3.68, df = 2265, *p* = 0.000). Additionally, the percentage of students with low health literacy was highest (Italy = 18.8%; Czechia = 17.7%; χ2 = 60.68, *p* = 0.000), and the percentage of students with high health literacy was the lowest (Italy = 7.8%; Germany = 12.8%; χ2 = 1 7.55, *p* = 0.000).

### 3.3. Sociodemographic, Family-, and School-Associated Factors

A hierarchical linear regression was run to test the three models. All assumptions for conducting multiple linear regression were met. The first model included sociodemographic variables and social stratifiers. No significant factors were found. The second model included the family variables. This model explains 12% of the variance in health literacy, with a significant increase over the previous model. Both paternal communication quality and family support were significant predictors. The third model included the school variables. This model explained an additional 55% of the variance. Both school satisfaction and school-related competence were significant predictors, while all other variables were not. The highest standardized coefficient was that of school-related competence, with a value of 0.23. Table 2 presents the three models.

## 4. Discussion

The WHO [33] has underlined the need to monitor health literacy in different population groups and has provided inputs for policies and intervention development and improvement. This issue is particularly important for adolescents because of the important effects of health literacy on their behaviours and well-being for both short and long periods of time. However, few comparative studies were developed about health literacy in different countries, and methodological improvement is necessary [37,42].

In this context, the Health Literacy for School-Aged Children scale (HLSAC) [41,43,44] is promising. It is a theory-based and comprehensive instrument to measure subjective health literacy among school-aged children. This scale was included in the HBSC protocol, providing the opportunity to monitor health literacy levels of representative samples of adolescents, to compare the results among countries and over time, and to verify the relationships between health literacy, social determinants, and health behaviours [28].

The first aim of this study was to assess the validity and reliability of the Italian version of the Health Literacy for School-Aged Children (HLSAC-I). Our results supported the validity of the one-factor model, which was preferred over the five-factor model. The fit indices were adequate but not excellent. However, our fit indices were very close to those obtained in other European countries (Poland χ2(df) = 168.83(35), *p* = 0.000; RMSEA = 0.08, CFI = 0.93, SRMR = 0.04; and Belgium χ2(df) = 69.23(35), *p* = 0.000; RMSEA = 0.07, CFI = 0.92, SRMR = 0.05) in a cross-national study [44]. This seems to indicate that while there are no specific problems in the Italian version of the HLSAC, there are possibly some minimal cultural differences from the other countries that showed a better fit of the model to data [44]. The Italian version of the HLSAC had a good level of reliability, with an ω of 0.80. All factor loadings were statistically significant, and item R^2^ was adequate. The only item with a low R^2^, “ability to follow the instructions given by doctors and nurses”, was one of the three items that was not invariant at the metric level in the cited cross-national study [44]. This is another indication of possible structural-cultural differences.

We are not aware of any other studies on the HLSAC that have adopted the WLSMV estimation method. Our analyses revealed that using the ML estimator with the four-step Likert scale of the HLSAC did not result in altered estimates or significantly different fit indices compared with the WLSMV estimator. This finding supports the validity of previous studies on HLSAC’s adoption of the ML estimation method.

The validation of the scale allowed for the identification of health literacy levels among Italian students. A total of 18.7% of the sample reported low levels, and only 6.8% reported high levels. Similar values were reported in all subgroups for sex, age or economic conditions. Italian 15-year-old students reported the lowest levels of health literacy compared with other countries [41].

This result highlights the importance of identifying the factors associated with health literacy and their possible determinants. The international literature has highlighted the influence of sociodemographic factors and social stratifiers [1,10,15,46,47,48]. Both family and school contexts were shown to have significant impacts. However, few studies have investigated the comprehensive effects of multiple contexts [53]. Moreover, variables related to the educational approaches and processes offered by families and schools remain understudied. This study aimed to identify the association between health literacy and multiple social determinants by considering the quality of relationships and the communication between students and relevant adults. A hierarchical linear regression was run, testing three models. Model 1 included sociodemographic characteristics and social stratifiers: sex, school level, mother’s and father’s educational level, and family economic condition. Model 2 added family characteristics to the previous variables: quality of maternal and paternal communication, and family support. Model 3 added certain school characteristics: school satisfaction, teacher support, and school-related competence and autonomy. The results showed no influence of sociodemographic variables or social stratifiers. However, this result does not confirm any previous study’s results [16]. It should be further investigated to account for the effects of these variables on health behaviours and other individual or contextual resources. The comparison between Models 2 and 3 provides interesting inputs. In Model 2, the quality of paternal communication and family support were significant predictors, although the beta values were quite low. However, when school variables were included in Model 3, these effects were no longer significant, and school satisfaction and school-related competence remained the only associated factors. Moreover, the increase in the variance of the third model is quite high, reaching 67% of the explained variance. An examination of the standardized coefficient revealed the importance of the school-related competence variable. This scale measures the ability of schools and teachers to promote student competence and autonomy in learning. These results confirm and empirically show the importance of promoting health literacy and health promotion in a school curriculum [2,18,28,29,30,31,32,33]. First, the results confirm the link between learning competencies and health skills. The school-related competence variable is focused on learning tasks and aims, but it was shown to be strongly associated with health literacy. Second, a school’s role in reducing inequalities is confirmed [8]. School connectedness and competence promotion are able to influence health literacy and overcome the effect of familial relationships. Third, these results show the relevance of an educational approach that is oriented to competence development and autonomic processes to promote connectedness with school and of the role teachers play as facilitators to promote health literacy [18,28,31]. Indeed, general teacher support was not significantly associated with health literacy.

### Limitations

This study presents some limitations. First, it included data collected in just one Italian region. Other studies should consider data from wider geographical areas. However, the use of HBSC data guaranteed a representative sample, and the Lombardy region is a very populated area with a number of inhabitants comparable to many European countries. Second, because this study did not test the measurement invariance of the Italian version of HBSC with those of the other European countries considered, the comparisons of Italian HBSC levels with other countries should be cautiously considered. The differences may be partly due to psychometric diversities. Specifically, we warned the reader to consider these results as a definitive indication that the level of health literacy among Italian 15-year-old students is the lowest of all the countries considered. However, given the results of another study testing measurement invariance among other European countries [41], if psychometric differences exist, these should not be extreme. Our comparisons can be safely interpreted as a rough indication of the level of health literacy among Italian students compared to that of other European countries. Third, the variables included for the family and school contexts are not exactly equivalent and measure different aspects of the connectedness and support that adults can provide to adolescents. However, all variables were indicators of the adults’ educational approaches in these contexts, particularly for connectedness and received support. Fourth, since the HBSC is a cross-sectional study, no inferences about causality can be defined. Longitudinal studies should verify the associations identified in this study.

## 5. Conclusions

This study has underscored the importance of monitoring adolescent health literacy. We validated the Italian version of the Health Literacy for School-Aged Children scale (HLSAC-I) [41,43,44] used in the HBSC study. In Italy, this scale was included only for the Lombardy region. Scale validation could be useful for other regions to monitor health literacy at the national level and compare the results among regions and countries. Moreover, this scale should be used to evaluate the effectiveness of health promotion programs. In addition, this instrument could be used to evaluate the effects of interprofessional education and collaborative practice on health literacy [34,35,36]. The descriptive data of the Italian sample showed lower levels of health literacy among 13- and 15-year-old students compared to other countries’ levels. This result highlights the need for health promotion policies targeted at adolescents. Finally, this study compared certain associated factors related to sociodemographic characteristics and social stratification, family, and school contexts. The results showed that school connectedness and educational approach are the most relevant factors. The study, therefore, confirms a school’s role in reducing inequalities and promoting health. Moreover, it shows that a whole-school approach that fosters school connectedness, promotes competence development, and encourages professional and organizational capacities is necessary to increase student health literacy. The Health Promoting School approach can be an effective strategy in this direction [18,28,31]. In the Lombardy region, this approach was developed over the last 10 years, and specific evidence-based programs to promote life skills were implemented on a large scale. This study contributes to the strengthening of these policies and the development and monitoring of health literacy.

## Figures and Tables

**Figure 1 ijerph-18-09943-f001:**
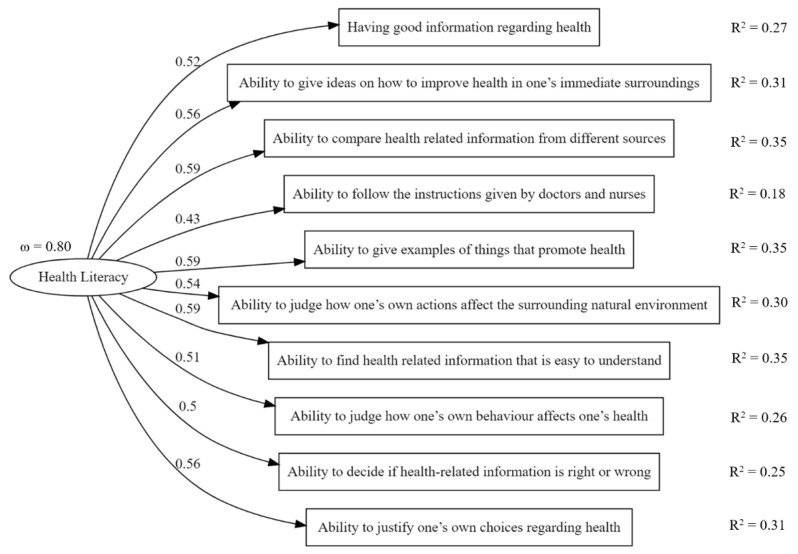
Standardized factor loadings, R^2^, and omega of the one-factor CFA of HLSAC-Italian version. All factor loadings were statistically significant (*p* < 0.001).

**Table 1 ijerph-18-09943-t001:** Sample health literacy levels.

	N (%)	Mean HL (SD)	Low HL	Moderate HL	High HL
Total	2148	29.08 (4.43)	18.7%	74.5%	6.8%
Female	1130 (53%)	29.17 (4.27)	17.2%	76.5%	6.4%
Male	1018 (47%)	28.98 (4.61)	20.3%	72.3%	7.4%
13 years old	1071 (50%)	29.05 (4.31)	18.5%	75.6%	5.9%
15 years old	1077 (50%)	29.11 (4.56)	18.8%	73.4%	7.8%
Low economic condition	457 (21%)	28.67 (4.39)	20.8%	73.7%	5.5%
Medium economic condition	1028 (48%)	29.22 (4.44)	17.5%	75.8%	6.7%
High economic condition	619 (29%)	29.12 (4.44)	18.7%	73.3%	7.9%

**Table 2 ijerph-18-09943-t002:** Hierarchical regression model.

Variables	Model 1			Model 2			Model 3		
	B (SE)	β	*p*	B (SE)	β	*p*	B (SE)	β	*p*
Sex	0.15 (0.21)	0.02	0.479	0.24 (0.21)	0.03	0.267	0.03 (0.21)	0.01	0.882
School level	−0.06 (0.21)	−0.01	0.787	0.05 (0.21)	0.01	0.802	0.52 (0.21)	0.06	0.014
Mother education	0.12 (0.12)	0.03	0.294	0.11 (0.12)	0.03	0.330	0.17 (0.11)	0.04	0.131
Father education	0.12 (0.11)	0.03	0.241	0.12 (0.10)	0.03	0.254	0.12 (0.10)	0.04	0.054
Economic condition	0.05 (0.05)	0.02	0.358	0.04 (0.05)	0.02	0.481	0.05 (0.05)	0.02	0.364
Maternal communication				0.09 (0.14)	0.02	0.491	0.05 (0.13)	0.01	0.692
Paternal communication				0.25 (0.12)	0.06	0.035	0.17 (0.11)	0.04	0.145
Family support				0.10 (0.09)	0.06	0.030	0.06 (0.09)	0.02	0.509
School satisfaction							0.33 (0.15)	0.06	0.031
Teachers support							0.13 (0.15)	0.02	0.378
School related competence							1.4 (0.16)	0.23	0.000
Corrected R^2^	0.002			0.012			0.067		
F for change in R^2^ (*p*)	1.628 (0.149)			7.032 (0.000)			35.486 (0.000)		

Dependent variable: health literacy. Sample size: *n* = 1769.

## Data Availability

The data underlying this article were provided by Lombardy Region by permission. Data will be shared on request to the corresponding author with permission of Lombardy Region.

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
