# Peer review of "Health Literacy Levels among Italian Students: Monitoring and Promotion at School"

_ijerph, 2021, doi:10.3390/ijerph18199943_

Round 1

Reviewer 1 Report

For the improvement of this manuscript, here are some comments  and suggestion :

â‘  Methods:

-I noticed that the original sample you included was 2287 , but in the statistical analysis part, the sample became 2148 . What is the principle of deleting the sample?The process of quality control needs to be described.

â‘¢ Results:

-Is the health literacy of 15-year-old adolescents in Italy comparable to that of other countries? I think this is a very important issue.Control other variables? How to control? Is this comparison rigorous?This part needs to be explained in detail.

Author Response

We thank you for your thoughtful suggestions and insights, which have helped us improved the paper significantly. We are also grateful for your positive feed-backs.

The manuscript has been rechecked and the necessary changes have been made in accordance with your suggestions. The responses to all comments have been prepared and given below.

  1. Method: I noticed that the original sample you included was 2287, but in the statistical analysis part, the sample became 2148. What is the principle of deleting the sample?The process of quality control needs to be described.

Response: Thanks for the very useful suggestion, which has helped us improved the paper significantly. The sample became 2148 because listwise deletion was deemed usable. We specified that the analyses were performed on a sample of 2148 students in the sample section and we explained the principle of deleting the sample in the analysis section.

  1. Results: Is the health literacy of 15-year-old adolescents in Italy comparable to that of other countries? I think this is a very important issue. Control other variables? How to control? Is this comparison rigorous?This part needs to be explained in detail.

Response: Thanks for the very useful suggestion, which has helped us improved the paper significantly. We specified in the analysis section the procedure used to compare the health literacy levels of 15-year-old adolescents in Italy with those of other countries. Moreover, we specified that no control variables were included in the present study or in the confronting study. Finally, we discussed the limits of this comparison in the limitations section.

Reviewer 2 Report

Thank you for the opportunity to perform the review. The paper is well written and clear and easy to understand. The study has three objectives: 1) to assess the validity and reliability of the Italian version of the Health Literacy for School-aged Children (HLSAC-I). 2) to identify the 1 health literacy levels among Italian students and compare them with other countries' levels, 3) to identify the associations between health literacy and multiple social determinants. The research questions fit the objective of the journal. The results are significant in regard to adolescent health literacy monitoring in Italy. A new finding is the validation results of the questionnaire and the use of the questionnaire also outside the Lombardy region in Italy. The research questions are addressed using adequate methods. The results are presented in a comprehensible manner. Discussion and conclusions are consistent with research question and results. The limitations of the study are adequately discussed.

The article addresses a relevant topic. The presentation of the methods and results is well structured and comprehensible for the reader.

Author Response

We thank you for your positive feed-backs.

Reviewer 3 Report

Thanks for submitting your paper to the International Journal of Environmental Research and Public Health. This is a very important topic especially in light of challenges posed by COVID-19 pandemic.

Please see my comments below –

Abstract – Very well written and provide necessary information about the study.

Introduction

1) Line numbers 36 – 36 – do not include any reference/in-text citations to support these statements. I would suggest that you include relevant in-text citations to support your points/thoughts.

2) Please revise this sentence  “Critical health literacy describes” (Line 41) – I would suggest critical health literacy includes..

3) Lines 47 – 59. I think authors have missed a very important piece of work/research when they address health literacy, health outcomes, long-term health conditions and COVID-19. Please review literature on interprofessional education and communication. It is important to include how health literacy can be a critical factor for effective inteprofessional communication. As researchers talk about COVID, I think it is important to look at this angle as this add value to literature review. I would encourage you to review work that has been completed in this field. Articles included below may help in building this section –

https://www.mdpi.com/2227-9032/9/5/567

https://ipe.asu.edu/practice/wesley-community-health-centers-addressing-social-determinants-health

https://library.iated.org/view/LEE2020IMP

4) Study aims – very clear and allows audiences to see what they are going to read in the rest of the paper.

Materials and Methods

1) I would encourage authors to include a table to describe the sample.

Results and Analysis – I do not have additional comments on this topic.

Discussion Section

I would suggest including interprofessional education especially with COVID - 19 pandemic. Health literacy is a very important topic especially for interprofessional communication. This will add value to the project.

Limitation and Conclusion sections are well written.

Author Response

We thank you for your thoughtful suggestions and insights, which have helped us improved the paper significantly. We are also grateful for your positive feed-backs.

The manuscript has been rechecked and the necessary changes have been made in accordance with your suggestions. The responses to all comments have been prepared and given below.

  1. Abstract – Very well written and provide necessary information about the study.

Response: We thank you for your positive feed-backs.

Introduction

  1. Line numbers 36 – 36 – do not include any reference/in-text citations to support these statements. I would suggest that you include relevant in-text citations to support your points/thoughts.

Response: Thanks for your useful suggestion. The references were reported before the sentence [1, 8-10], but they were unclear. We moved the references after the lines you commented.

  1. Please revise this sentence  “Critical health literacy describes” (Line 41) – I would suggest critical health literacy includes.

Response: Thanks for your useful suggestion. We changed the sentences as suggested.

  1. Lines 47 – 59. I think authors have missed a very important piece of work/research when they address health literacy, health outcomes, long-term health conditions and COVID-19. Please review literature on interprofessional education and communication. It is important to include how health literacy can be a critical factor for effective inteprofessional communication. As researchers talk about COVID, I think it is important to look at this angle as this add value to literature review. I would encourage you to review work that has been completed in this field. Articles included below may help in building this section –

https://www.mdpi.com/2227-9032/9/5/567

https://ipe.asu.edu/practice/wesley-community-health-centers-addressing-social-determinants-health

https://library.iated.org/view/LEE2020IMP

Response: Thanks for your useful suggestion and for highlighting a very important piece of work/research. We introduce the links between health literacy and interprofessional education, considering the school's role in this issue. Actually, the school can have a crucial role in offering interprofessional education and collaborative practice.

  1. Study aims – very clear and allows audiences to see what they are going to read in the rest of the paper.

Response: We thank you for your positive feed-backs.

Materials and Methods

  1. I would encourage authors to include a table to describe the sample.

Response: Thanks for the very useful suggestion, which has helped us improved the paper significantly. The sample characteristics were already reported in the Table 1. However, the reference to the table was missing in the sample section. We added it.

  1. Results and Analysis – I do not have additional comments on this topic.

Response: We thank you for your positive feed-backs.

Discussion Section

  1. I would suggest including interprofessional education especially with COVID - 19 pandemic. Health literacy is a very important topic especially for interprofessional communication. This will add value to the project.

Response: Thanks for your useful suggestion. In the conclusion section, we added a study implication related to the links between interprofessional education and collaborative practice and health literacy.

  1. Limitation and Conclusion sections are well written.

Response: We thank you for your positive feed-backs.